# Opposing Roles of S1P_3_ Receptors in Myocardial Function

**DOI:** 10.3390/cells9081770

**Published:** 2020-07-24

**Authors:** Dina Wafa, Nóra Koch, Janka Kovács, Margit Kerék, Richard L. Proia, Gábor J. Tigyi, Zoltán Benyó, Zsuzsanna Miklós

**Affiliations:** 1Institute of Translational Medicine, Semmelweis University, 1094 Budapest, Hungary; kochnori@gmail.com (N.K.); kovacsjankee@gmail.com (J.K.); margit.nagy9@gmail.com (M.K.); gtigyi@uthsc.edu (G.J.T.); benyo.zoltan@med.semmelweis-univ.hu (Z.B.); 2National Institute of Diabetes and Digestive and Kidney Diseases (NIDDK), National Institues of Health, Bethesda, MD 20892, USA; richard.proia@nih.gov; 3Department of Physiology, University of Tennessee Health Science Center, Memphis, TN 38163, USA

**Keywords:** sphingosine-1-phosphate, ischemia/reperfusion, cardioprotection, vasoconstriction, coronary flow, myocardial function, myocardial infarct, albumin

## Abstract

Sphingosine-1-phosphate (S1P) is a lysophospholipid mediator with diverse biological function mediated by S1P_1–5_ receptors. Whereas S1P was shown to protect the heart against ischemia/reperfusion (I/R) injury, other studies highlighted its vasoconstrictor effects. We aimed to separate the beneficial and potentially deleterious cardiac effects of S1P during I/R and identify the signaling pathways involved. Wild type (WT), S1P_2_-KO and S1P_3_-KO Langendorff-perfused murine hearts were exposed to intravascular S1P, I/R, or both. S1P induced a 45% decrease of coronary flow (CF) in WT-hearts. The presence of S1P-chaperon albumin did not modify this effect. CF reduction diminished in S1P_3_-KO but not in S1P_2_-KO hearts, indicating that in our model S1P_3_ mediates coronary vasoconstriction. In I/R experiments, S1P_3_ deficiency had no influence on postischemic CF but diminished functional recovery and increased infarct size, indicating a cardioprotective effect of S1P_3_. Preischemic S1P exposure resulted in a substantial reduction of postischemic CF and cardiac performance and increased the infarcted area. Although S1P_3_ deficiency increased postischemic CF, this failed to improve cardiac performance. These results indicate a dual role of S1P_3_ involving a direct protective action on the myocardium and a cardiosuppressive effect due to coronary vasoconstriction. In acute coronary syndrome when S1P may be released abundantly, intravascular and myocardial S1P production might have competing influences on myocardial function via activation of S1P_3_ receptors.

## 1. Introduction

Ischemic heart disease, including acute coronary syndrome (ACS), is a major cause of death worldwide [1]. ACS is the sudden loss of adequate blood perfusion to the heart, most commonly initiated by the rupture of an atherosclerotic plaque and consequent activation of blood coagulation. This process results in thrombotic occlusion of the coronary artery causing cardiac tissue damage [2]. Urgent reestablishment of blood perfusion to the affected area is crucial to minimizing ischemic tissue injury. Besides the therapeutic time window, the success of reperfusion depends on several other factors such as vascular response to pathophysiological events happening prior to and during thrombus formation. Platelet activation might be relevant in this context as it releases numerous vasoactive mediators which might have an impact on the dynamics and severity of ischemic injury. Sphingosine-1-phosphate (S1P) is one of these many mediators [3,4,5,6,7,8].

S1P is a sphingolipid mediator which is produced by a wide variety of cells [9]. In vivo, albumin and APO-M in HDL are the most recognized carriers of S1P in blood plasma, which have also been reported to modulate the actions of S1P [10,11]. S1P actions include regulation of diverse physiological and pathophysiological processes such as inflammation, autoimmunity, and neurodegeneration [12,13]. In the cardiovascular (CV) system, activated platelets synthesize and release S1P in large amounts [3,4,5,6,7,8], and S1P has been reported to play a role in regulating vascular tone [14,15], atherogenesis, cardiac remodeling, and cardioprotection [16,17,18]. To date, five different G protein-coupled receptors belonging to the endothelial differentiation gene (EDG) family have been identified as specific S1P receptors (S1P_1–5_) [19,20]. From these, S1P_1_, S1P_2_ and S1P_3_ receptors are expressed abundantly in the CV system and have been reported to mediate CV actions of S1P [16].

S1P has been attributed with cardioprotective effects against ischemia-reperfusion (I/R) injury by several research groups [18,21,22,23,24,25,26,27]. The key enzymes in S1P synthesis, sphingosine-kinase 1 and 2 (SphK1 and 2), have been implicated in the ischemia-induced increased release of S1P from cardiomyocytes as well as in mediating the beneficial effects of ischemic pre- and post-conditioning [18,23,24]. Combined deletion of S1P_2_ and S1P_3_ receptors increased the infarcted area and enhanced apoptotic cell death after I/R, suggesting that activation of these receptors is cardioprotective [25].

Preischemic S1P treatment has also been reported to decrease infarct size in ex vivo experimental models [24,26]. It has already been shown by Theilmeier and colleagues that HDL and S1P directly protect the heart against I/R injury via the S1P_3_ receptor in vivo [27]. However, in ACS when S1P is released in substantial amounts from platelets and endothelial cells in blood plasma, it might bind to carriers other than HDL. Several studies have highlighted the vasoconstrictive effects of S1P in various vessel beds from multiple species including the coronaries: S1P had a constrictive effect on isolated porcine pulmonary artery rings [28], in canine, rat, murine, and leporine basilar and middle cerebral arteries [29,30], in rat portal veins [31] and in canine coronaries [32]. S1P administration to the coronary perfusate has been shown to diminish coronary flow (CF) in Langendorff-perfused rat hearts [33]. This effect was attenuated by pharmacological inhibition of S1P_3_ receptors in the same experiment [33]. Another study conducted on coronary smooth muscle cells raised the potential involvement of S1P_2_ receptors in the vasoconstrictor response elicited by S1P [34]. Beside its actions on CF, S1P exerts other short-term cardiac effects including generation of arrhythmias and negative inotropy [35,36,37,38].

The cardiac effects of S1P reported in I/R injury are controversial. Activation of S1P receptors seems to be cardioprotective, whereas the acute effects of S1P to reduce CF and cardiac contractility are expected to interfere with successful post-ischemic recovery. Moreover, S1P_2_ and S1P_3_ receptors have been shown to be involved in both mechanisms. In ACS, when S1P is released in large amounts from activated platelets, its favorable and potentially deleterious effects might clash with one another. In the present study, we aim to delineate how these opposing S1P actions actually affect postischemic cardiac injury after a non-fatal ischemic insult.

For this purpose, we conducted ex vivo experiments in isolated murine hearts mounted on the Langendorff-system. First, we mimicked ACS-related massive S1P release into the blood in order to characterize its coronary effects and consequences on heart function. In these experiments we used albumin as an S1P chaperone and also in subsequent experiments Krebs buffer without added chaperone protein as the vehicle. Second, using S1P receptor gene knockout (KO) mouse models, we aimed to identify receptors involved in these cardiac effects. Third, to understand the role of S1P_3_ receptor in cardioprotection, we applied a non-fatal I/R protocol in S1P_3_ deficient hearts. Finally, the complete sequence of ACS was modeled with an initial exposure of the coronaries to S1P as it occurs during plaque rupture and platelet activation, followed by 20 min of complete ischemia, during which the myocardial S1P-producing machinery can be activated, and concluding with 120 min reperfusion representing successful reopening of the coronary artery in a clinical setting. With this approach, we were able to separate the consequences of intravascular and myocardial S1P-related effects during ACS and also to evaluate their combined effects.

## 2. Materials and Methods

### 2.1. Animals

All experiments reported here were performed in hearts of 130–150-day-old male mice. Animals were bred and housed in the animal facility at Semmelweis University, kept in a 12/12-h light/dark cycle and with free access to water and food. C57BL/6 (WT) mice were bred from breeding pairs obtained from Charles River Laboratories (Isaszeg, Hungary). To answer our specific questions, S1P_2_-KO and S1P_3_-KO animals along with wild-type littermates on C57BL/6 genetic background were tested [39]. All procedures were carried out according to the guidelines of the Hungarian Law of Animal Protection (28/1998) and were approved by the Government Office of Pest County (Permission number: PEI/001/820-2/2015).

### 2.2. Isolated Perfused Heart Experiments

General anesthesia was induced by intraperitoneal injection of 40 mg/kg pentobarbital, followed by thoracotomy and isolation of the heart. The isolated heart was mounted in a Langendorff apparatus (Experimetria Ltd., Budapest) and perfused at constant 80 mmHg pressure with modified Krebs-Henseleit buffer (118 mM NaCl, 4.3 mM KCl, 25 mM NaHCO_3_, 1.2 mM MgSO_4_, 1.2 mM KH_2_PO_4_, 0.5 mM NaEDTA, 2.0 mM CaCl_2_, 11 mM glucose, 5 mM pyruvate (pH 7.4) - all purchased from Sigma-Aldrich, Budapest, Hungary) [40,41,42,43]. The solution was continuously gassed with 95% O_2_ and 5% CO_2_ at 37 °C. During the experiment, the heart was surrounded by a thermally-regulated chamber filled with Krebs-Henseleit buffer.

CF was continuously monitored with a transit-time flow meter placed into the inflow line (Transonic 2PXN flow probe, Transonic Systems Inc., Ithaca, NY, USA). In order to measure left ventricular pressure (LVP), a fluid-filled balloon catheter connected to a pressure transducer was inserted into the ventricle to maintain diastolic pressure at 8 mmHg.

Devices were connected to a computer and data were recorded and analyzed by the Haemosys software (Experimetria Ltd., Budapest, Hungary). Left ventricular developed pressure (LVDevP) was calculated as the difference between peak systolic and minimum diastolic (LVDiastP) pressures. The positive and negative maximum values of the first derivative of the LVP (+dLVP/dt_max_, −dLVP/dt_max_) were determined as indices of left ventricular contractile and lusitropic performance, respectively.

### 2.3. Experimental Protocol

After cannulation of the isolated heart, a 30-min stabilization period was allowed. Subsequently, baseline data were recorded and S1P (D-erythro-sphingosine-1-phosphate, Avanti Inc., 10^−6^ M) or vehicle was infused to the perfusion line for 5 min. One mg S1P was dissolved in 263 µL 0.3 N NaOH. This solution (10^−2^ M) was further diluted with Krebs solution to give the required concentration in the perfusate.

First, to characterize S1P effects on CF we performed dose-response experiments with S1P in the concentration range of 10^−9^–10^−5^ M. S1P was administered in the presence of S1P chaperon human serum albumin (HSA) (Sigma Aldrich, Budapest, Hungary, Cat.No: A3782, lyophilized powder, fatty acid free, globulin free, >99%) or diluted directly in chaperone-free Krebs buffer. The molar S1P to HSA ratio was 1:2 at every concentration [44]. In these experiments we applied S1P in cumulative doses, the next S1P dose was added to the previous dose after the response had reached its maximum. The biological effects of the applied vehicles were tested in separate experiments, and they proved to be without an effect.

In order to answer our main questions, two different experimental protocols were tested (Appendix A). To understand the effects of S1P under stable baseline conditions, a 5-min S1P (or vehicle) infusion was applied that was followed by a 20-min washout period. Whereas in the I/R-injury protocol, the 5-min S1P (or vehicle) infusion was followed by a 20-min global ischemia that was brought about by complete cessation of perfusion. At the end of the ischemic period, perfusion was restarted and reperfusion was maintained for 2 h. In these protocols, S1P was delivered in albumin free Krebs solution at a concentration of 10^−6^ M. The S1P concentration applied was defined as a dose approximating its ED_50_ value (1.17 * 10^−6^ M in Krebs solution).

### 2.4. Measurement of Infarct Size

In the I/R experiments, hearts were removed from the apparatus after the 2-h reperfusion period and placed into a −20° C freezer for at least 15 min. The left ventricle of the frozen heart was cut into ~1 mm thick slices (4 to 6 slices per heart). To visualize the infarcted area of the heart, the slices were then incubated in a phosphate buffer containing 1% triphenyltetrazolium (TTC) (Sigma-Aldrich) for 20 min at a temperature of 37 °C. The TTC powder was diluted in a two-part phosphate buffer system at a pH of 7.4 and the slices were fixed in 10% formalin for 15 min [45]. Photos of the TTC-stained slices were captured using a stereomicroscope equipped with a high-resolution digital camera (Rasband, W.S.) and analyzed using Image-J software (National Institutes of Health, Bethesda, MD, USA). The area at risk, defined as the total area consisting of the pale plus red parts and the infarcted pale area, was measured and relative infarct size was calculated as a percentage of the area at risk.

### 2.5. Statistical Analysis

Results are presented as mean ± standard error of the mean (SEM). In order to compare time series data between 2 experimental groups, we used two-way repeated measurement ANOVA and Dunnett’s *post hoc* test for multiple comparisons. Comparison of data acquired from the 4 experimental groups (WT/S1P_3_-KO vs. vehicle/S1P infusion) was performed by two-way ANOVA and Sidak’s *post hoc* test. To compare the maximal effects of S1P infusion, unpaired *t*-test was applied. To determine the total perfusion loss during S1P infusion, area over the curve (AOC) was calculated. In dose-response experiments non-linear regression analysis was used to find the best fit and ED_50_ values. To compare dose-responses between 2 experimental groups, comparison of Fits analysis of the statistical software was applied. All statistical analyses were performed using GraphPad Prism 7.0 (San Diego, CA, USA) and *p* < 0.05 was considered as statistically significant.

## 3. Results

### 3.1. Dose-Dependent Effects of Intravascular S1P on CF Administered with or without S1P-Chaperon Albumin

To characterize the effect of intravascular S1P on CF, we carried out dose-response experiments with and without albumin as a chaperone. When administered without a carrier, S1P elicited a dose-dependent CF reduction in isolated hearts with an ED_50_ value of 1.17 × 10^−6^ M. Therefore, the S1P in further experiments was applied at 1 microM—a dose close to its ED_50_ value. The coronary effect of S1P was similar in the presence of albumin, however the ED_50_ value slightly shifted to a smaller concentration range (1.85 × 10^−7^ M) though it was not statistically significant (*p* = 0.12, *F* = 2.46) (Figure 1). The maximal reduction in CF was also indistinguishable between groups regardless whether S1P was applied carrier-free or with albumin as vehicle.

### 3.2. Effects of Intravascular S1P Exposition on CF and Heart Function

To investigate the effects of a robust S1P release on CF and cardiac function, 10^−6^ M S1P or its vehicle was administered to the perfusate of isolated WT murine hearts for 5 min. Administration of S1P reduced CF by 44 ± 3% (Figure 2A). This remarkable decrease started at the beginning of the S1P infusion and continued progressively during the 5 min. During the 20-min wash-out period, CF did not return to the baseline level and remained at a significantly lower value (*p* < 0.0001).

CF reduction induced by S1P coexisted with compromised left ventricular contractile performance, which is indicated by a 54 ± 9% drop in LVDevP (Figure 2B) and by the markedly decreased +dLVP/dt_max,_ and −dLVP/dt_max_ values (*p* < 0.0001) (Figure 2C–D). The vehicle did not affect either CF or other measured heart function parameters (Figure 2A–D).

Earlier studies suggested that S1P might affect coronaries via S1P_2_ and S1P_3_ receptors [33,34]. Therefore, we aimed to identify which of these receptors mediate(s) the effect of S1P on the CF. For this purpose, we perfused S1P into the isolated hearts of S1P_2_ (Figure 3) and S1P_3_ (Figure 4) KO mice following the experimental protocol described above.

The CF-reducing effect of S1P developing in S1P_2_-deficient mice was similar to that of WT littermates (Figure 3A–C). The drop of the LVDevP was also similar in the two groups (Figure 3D–F), with no statistically significant difference.

In S1P_3_-KO hearts, the CF-reducing effect of S1P was markedly diminished compared to WT mice (Figure 4A). There was a significant difference in the maximal effects: CF was dropped by 1.95 ± 0.33 mL/min in WT and only by 0.93 ± 0.10 mL/min in S1P_3_-KO mice (Figure 4B). The AOC used as an index for total perfusion loss during the infusion period showed similar decrease. During the 5-min S1P infusion, the total perfusion loss was 8.56 ± 1.60 mL in WT vs. 3.70 ± 0.57 mL in S1P_3_-KO mice (Figure 4C).

The decrease in left ventricular contractile performance upon S1P infusion was also attenuated in S1P_3_-KO mice (Figure 4D): both the maximal drop in LVDevP (Figure 4E) and the area over the LVDevP curve used as a measure of loss of contractile activity (Figure 4F) were significantly reduced compared to WT controls.

### 3.3. Role of Myocardial S1P_3_ Receptor Activation in I/R Injury

To better understand the apparent contradiction between the widely reported cardioprotective and observed cardiosuppressive effects of S1P, we aimed to separate the myocardial and coronary actions of S1P in a model of I/R injury.

First, we investigated the effects of potential S1P_3_ receptor activation during I/R in the absence of intravascularly administered S1P. WT and S1P_3_-KO hearts were exposed to an I/R protocol, CF and myocardial function were monitored during reperfusion. CF did not differ significantly between the WT and S1P_3_-KO mice (Figure 5A1). In contrast, parameters describing myocardial performance showed marked differences. The lack of S1P_3_ resulted in a far worse postischemic functional recovery as evidenced by the drop of the LVDevP (Figure 5B1), decreased +dLVP/dt_max,_ and −dLVP/dt_max_ (Figure 5C1,D1), and elevated LVDiastP (Figure 5E1).

These results indicate that S1P_3_ receptors play a beneficial role in preventing ischemia-induced myocardial dysfunction, most probably by activation from S1P generated locally by the tissues of the ischemic heart. However, this myocardial S1P release did not induce S1P_3_-mediated coronary vasoconstriction as observed in the previous experiments with intravascular S1P administration.

Interestingly, lack of S1P_2_ receptors did not influence any of these functional parameters nor the infarct size in this I/R model (Appendix A.)

### 3.4. Effects of Preischemic Intravascular S1P Exposure on I/R Injury

Next, we investigated the role of intravascular S1P by administering S1P to the perfusion solution before I/R at a concentration of 10^−6^ M for 5 min. Under these conditions, CF returned to a significantly higher value during reperfusion in the S1P_3_-KO hearts (Figure 5A2) indicating S1P_3_-mediated coronary vasoconstriction. Postischemic myocardial function failed to return during the reperfusion without any difference between the two groups (Figure 5B2–E2).

In order to determine the effects of S1P infusion on postischemic CF and cardiac performance, results obtained by the two experimental protocols were compared (see the table panels in Figure 4). S1P-exposed WT hearts showed a marked reduction in post-ischemic functional myocardial recovery as compared to WT or S1P_3_-KO hearts without S1P administration (Figure 5B3–E3). Furthermore, the difference in the functional parameters between WT and S1P_3_-KO hearts was not statistically significant (B2–E2 table insets in Figure 5.).

Finally, we determined whether alterations in myocardial function were reflected in the irreversible ischemic damage of cardiomyocytes. TTC staining revealed that without S1P administration, the relative infarct size was larger in S1P_3_-KO (10.72 ± 2.93%) than in WT (1.12 ± 0.37%) hearts (Figure 6A,C). In the S1P-exposed groups, the infarcts were substantially larger, but they did not differ between S1P_3_-KO and WT hearts (Figure 6B,D).

Comparing the size of the infarcted area in S1P-pretreated (Figure 6B,D) with the untreated groups (Figure 5A,C), we detected a marked increase in the size of the infarcted myocardium as a result of S1P administration (Figure 6E).

## 4. Discussion

In the present ex vivo study, three different experimental protocols were tested in order to understand the complexity of S1P-induced alterations of cardiac function in ACS and also to determine the involvement of S1P receptor subtypes in mediating these effects. First, we focused on intravascular S1P release, which occurs at the onset of ACS when plaque rupture initiates platelet activation. We found that S1P caused a 44 ± 3% reduction of heart perfusion and simultaneous suppression of myocardial contractility (54 ± 9% decrease in LVDevP). Both effects were attenuated in hearts of S1P_3_-KO mice, indicating a major role of S1P_3_ in signaling. In the second part of our study, we focused on the effects of S1P receptor activation within the heart in response to ischemia. Under these conditions, hearts of S1P_3_-KO mice exhibited worse postischemic contractile recovery (82 ± 3% vs. 52 ± 3% decrease in LVDevP compared to preischemic baseline value in S1P_3_-KO and WT mice, respectively) and larger infarct size (11 ± 3% vs. 1 ± 3% in S1P_3_-KO and WT mice, respectively) than WT hearts, indicating that ischemia-induced myocardium-related S1P actions are cardioprotective via activation of S1P_3_. Finally, we proposed to model the complex scenario of ACS, when intravascular and myocardial S1P release may occur simultaneously and influence cardiac function. Under these conditions, WT hearts showed limited coronary perfusion without any sign of postischemic functional recovery. In S1P_3_-KO hearts, coronary reflow was better (Figure 5A2), but this failed to improve cardiac function (Figure 5B2–E2) or to reduce infarct size (Figure 6B) compared to WT. These observations indicate that although S1P_3_-mediated vasoconstriction contributes to the deleterious no-reflow phenomenon, elimination of this effect in S1P_3_-KO hearts does not moderate I/R injury because it also abolishes the benefits of S1P_3_-mediated cardioprotection.

The major sources of S1P in blood plasma are red blood cells, platelets, and endothelial cells [5,9]. Sphingosine kinase is highly active in platelets and synthesizes S1P from sphingosine taken up from plasma and produces it in the outer leaflet of the platelet plasma membrane [4]. Platelets store S1P abundantly and release it upon activation [3,7,8]. In ACS, when blood clotting is activated by the rupture of an atherosclerotic plaque, substantial amount of S1P might be released to the circulation [6]. S1P has been reported to have vasoconstrictor and endothelium-dependent vasodilator actions in different vascular beds [14,46,47]. For instance, S1P was shown to have a constrictor effect in isolated porcine pulmonary artery rings [28], in canine, rat, murine and leporine basilar and middle cerebral arteries [29,30], in rat portal veins [31], and in canine coronaries [32]. Whereas S1P increased NO production in cultured HUVEC [48] and in bovine lung microvascular endothelial cells [49], and HDL, a carrier of S1P was shown to cause endothelium-dependent vasodilation in aortic rings of rats and mice mediated via S1P_3_ activation [50]. However, despite its potential pathophysiological relevance, only a few of these studies have investigated the effects of S1P on the coronaries, and none of them have attempted to relate it to myocardial function. In our study, we found that S1P causes dose-dependent reduction in CF of Langendorff-perfused murine hearts. This observation is in agreement with earlier reports that also ascribed vasoconstrictor effects to S1P in the coronaries and other vascular beds [33,34,51]. Murakami et al. reported dose-dependent S1P-induced CF reduction in rat hearts in a similar experimental model [33]. When we delivered S1P in the presence of albumin, the coronary effect of S1P remained unchanged (Figure 1.). Although, a slight, nonsignificant shift in the ED_50_ to lower concentration was observed potentially indicating that albumin may enhance S1P coronary effects by protecting it from degradation by phosphatases in the vessels of isolated hearts.

One of the main aims of our study was to (1) mimic the effect of robust S1P exposure of the coronary arteries that might occur in ACS upon thrombotic platelet activation, and (2) explore its effects on coronary perfusion, and (3) on heart function. For this purpose, we administered S1P to the coronary perfusate of isolated murine hearts at 1 microM, a concentration that might easily occur in a thrombotic coronary artery [5,6,52,53], and was close to the ED_50_ value, we defined (Figure 1). This produced a remarkable decrease in CF (Figure 2A). This observation is in agreement with earlier studies, which also ascribed vasoconstrictor effects to S1P in coronaries and other vascular beds [33,34,51]. Murakami et al. reported dose-dependent S1P-induced CF reduction in rat hearts in a similar experimental setting [33]. The S1P-induced flow deprivation in our study was associated with a significant decline in cardiac performance, which was evidenced by decreased LVDevP, +dLVP/dt_max_ and −dLVP/dt_max_ (Figure 2B–D). This might be primarily attributed to CF reduction. However, direct negative inotropic effect of S1P on cardiomyocytes reported by earlier studies might also play a role [25]).

The cellular actions of S1P are attributed to the presence of five specific G protein-coupled S1P receptors [19,20]. Among these, S1P_1_, S1P_2_, and S1P_3_ receptors are expressed abundantly in the CV system [16]. Detailed description of S1P signaling in coronaries is not available in the literature. However, a few studies provide evidence that S1P_2_ or S1P_3_ might play a role in the regulation of heart function. In a recent study, we reported a dominant role of S1P_2_ in S1P-induced enhancement of vasoconstrictor stimuli in the circulation [54]. Therefore, in the present study we aimed to characterize the role of these two receptors in mediating CF reduction by S1P. Using an S1P_3_-KO mouse model, we showed that the S1P_3_ receptor plays a relevant role in mediating S1P-induced CF reduction, because the absence of this receptor diminishes significantly the CF reducing effect of S1P (Figure 3C). This observation confirms the findings of Murakami *et al*., who proposed the role of S1P_3_ in coronary constriction using the S1P_3_ receptor antagonist TY-52156 in a similar experimental setting [33]. Levkau et al. found that S1P decreases myocardial perfusion in vivo and this effect was absent in S1P_3_-KO mice [51]. Other investigators proposed the role of S1P_2_ receptors because S1P induced constriction in human coronary smooth muscle cells that was attenuated by the S1P_2_ antagonist, JTE-013 [34]. However, in our experiments S1P_2_-KO mice did not reproduce these pharmacological observations. We acknowledge that this does not necessarily mean that the S1P_2_ has no role in regulating coronary vessel tone, because it might be that S1P_2_ also activates pathways in the heart which cause coronary dilation, and these and the direct vasoconstrictor effects in smooth muscle cells canceled out each other in our experiments. However, this putative mechanism requires further investigation. Moreover, S1P_2_ activation might also sensitize the smooth muscle to other vasoconstrictor stimuli, as has been shown in the systemic circulation [54].

S1P is frequently implicated in cardioprotection [21,22,55,56]. Indeed, numerous studies have shown that it decreases the infarcted area and apoptotic cell death after I/R injury, and that it plays a role in the mechanism of ischemic pre- and post-conditioning [18,23,24,25,26,27]. However, myocardial function has not yet been evaluated in detail in these previous studies, although the involvement of S1P_2_ and S1P_3_ receptors has already been suggested [27,33,34]. This protective effect has been inferred from experiments, through the use of fundamentally different methodological approaches. In most of these studies, inhibition of S1P signaling in ischemia was achieved by using S1P receptor gene-deficient models or the pharmacological inhibition of SphK1 and SphK2 enzymes, which made I/R injury more severe and/or reduced the benefits of ischemic pre- and post-conditioning. These observations suggest that S1P signaling is stimulated in ischemia most likely by locally generated S1P released from the heart tissue.

The other approach introduced intravascular S1P administration into the coronary blood flow before an ischemic insult. Although this experimental setting can be considered as a relevant model for studying S1P effects in ACS, inasmuch as S1P infusion mimics S1P release during thrombus formation, whereas the flow cessation models thrombotic occlusion, only a few investigators have explored S1P effects this way, and they only assessed tissue damage without monitoring postischemic heart function. Nevertheless, these studies consistently reported a decrease in the infarcted area [24,27]. This is surprising considering that S1P has several short-term effects in the heart by reducing CF and causing negative inotropy that might be detrimental to postischemic contractile recovery [35,36,37,38].

Our current study was designed to combine these approaches in the context of S1P receptor signaling. Our choice of focus on S1P_3_ signaling was motivated by the results of our experiments shown in Figure 3 and Figure 4 which highlight that the short-term cardiac effects of S1P are mediated in large part by S1P_3_. However, S1P_2,_ which is the other receptor proposed to participate in cardioprotection [25], did not have a major role. Moreover, the exposure of S1P_2_-KO hearts to our I/R protocol produced similar functional and tissue injury to that observed in control hearts (Appendix A), showing that this receptor has no detectable role in cardioprotection in our experimental setting.

First, we aimed to clarify whether intrinsic activation of S1P_3_ signaling during ischemia was protective in our experimental setting. Our results showed that, in the absence of S1P_3,_ murine hearts were more susceptible to a 20-min global ischemia. This was indicated by weaker contractile recovery during the 2-h reperfusion period (Figure 5B1–D1), higher postischemic end-diastolic pressure (Figure 5E1) an indicator of more severe myocardial ischemic contracture, and increased infarct size (Figure 6A,C). These observations are in agreement with other studies which also implicated the participation of S1P_3_ signaling in cardioprotection against I/R injury [25,26,27]. Notably, the severe functional and morphological injury in S1P_3_-KO hearts developed despite a relatively maintained CF, which approached the preischemic value and was not worse than that of WT hearts during the reperfusion period (Figure 5A1). The observation that CF during reperfusion was similar in WT and S1P_3_-KO hearts indicates that vascular S1P_3_ was not exposed to S1P levels sufficient to induce S1P_3_-mediated vasoconstriction. This is not surprising if we consider that the perfusion fluid was free of exogenously added S1P.

In our study we also investigated the effects of S1P on I/R injury by the other approach described in the literature, where S1P was administered to the coronary circulation before ischemia. After S1P pretreatment, we applied a non-fatal ischemia protocol and followed up the recovery of cardiac function upon reperfusion. This model allowed for the exploration of S1P actions which may take place in a complex ACS scenario in which simultaneous intravascular and myocardial S1P release may occur. Preischemic S1P infusion can be considered as simulation of platelet-derived S1P release in ACS [3,4,5,6,7,8], whereas the ischemia protocol as S1P release from the myocardium [23,24]. Furthermore, our experiments investigated S1P effects more broadly than previous studies. In addition to determining infarct size, we also assessed postischemic cardiac function. We found that preischemic S1P exposure exacerbated ischemic injury. After an ischemic insult which is supposed to be non-fatal, the infarcted tissue extended to a large part of the myocardium and restitution of contractile activity was hardly observed in WT hearts. The latter was indicated by extremely low LVDevP, +dLVP/dt_max_, and −dLVP/dt_max_ values (Figure 5) that failed to approach preischemic levels during reperfusion, although CF partly recovered. Comparing ischemic injury of S1P pretreated and non-treated WT hearts, infarct size was significantly larger (Figure 6E), whereas LVDevP, +dLVP/dt_max_ and - dLVP/dt_max_ (Figure 5B3–D3) values were significantly lower at the end of the 2-h reperfusion period. Interestingly, some researchers observed a decrease in infarct size after preischemic S1P treatment [24,26,27]. This difference might be explained by differences in methodology because the infusion time and concentration of S1P preinfusion were slightly different [21,24,26]. A possible explanation can be that our infusion protocol, where we applied S1P at 10^−6^ M, might have caused sustained desensitization of cardiomyocytes to S1P. Several studies have shown in different cell types that S1P causes rapid desensitization of S1P receptors which persists for hours [57,58]. It might well be that in our experimental setting the combination of sustained vasoconstriction, which is potentially detrimental to postischemic recovery, and the loss of S1P_3_ mediated activation of prosurvival and antiapoptotic protective pathways due to desensitization, results in enhanced I/R injury. Whereas, in experimental settings, where lower doses are used (0.1 micromolar), S1P_3_ mediated protection is more active and dominates over effects of more moderate perfusion loss.

Because we observed that the coronary effects of S1P are in part mediated by S1P_3_ (Figure 4), we also explored the effects of preischemic S1P exposure on ischemic damage in S1P_3_-KO hearts. Although preischemic CF and the function of S1P_3_-KO hearts were better (data not shown, also cf. Figure 4), their functional recovery was as weak, and infarct size as large, as those of WT hearts. Interestingly, although CF in S1P_3_-KO hearts returned close to the preischemic value, this relatively better perfusion did not provide any benefit for cardiac performance. All these results indicate, that although the absence of S1P_3_ receptors might mitigate the detrimental effects of preischemic intravascular S1P exposure by decreasing the CF-reducing effect of S1P and allow for better reflow during reperfusion, the concomitant loss of S1P_3_-mediated cardioprotection obliterates this potential benefit. Therefore, the S1P_3_ receptor seems to mediate two opposing S1P actions in the heart, as schematically shown in Figure 7.

## 5. Conclusions

In this study, using isolated perfused murine hearts, we designed experimental models to simulate and explore the actions of S1P release in ACS. First, we described the effects of intravascular S1P exposure which occur during thrombus formation; and second, the effects of S1P within the heart in response to myocardial ischemia in a separate experimental paradigm. Finally, we investigated the combined effects of simultaneous intravascular and myocardial S1P effects, which can be a real-life scenario in ACS. Intracoronary administration of S1P caused a substantial decrease in CF and heart function. Using S1P receptor KO mouse models, we established that S1P_2_ has only a minor role, whereas S1P_3_ is a key determinant in this effect. In I/R experiments, the postischemic functional recovery was weakened and the ratio of the infarcted area was increased in S1P_3_-KO hearts, confirming the cardioprotective role of this receptor subtype. Preischemic intravascular S1P administration worsened the recovery of cardiac function and increased infarct size both in WT and S1P_3_-KO hearts, although coronary hypoperfusion was attenuated by S1P_3_ deficiency. These findings highlight that S1P has opposing effects in the myocardium: S1P released from the ischemic myocardium seems to be cardioprotective, whereas S1P acting via the coronary circulation is deleterious to the heart. Moreover, both of these effects are in a large part mediated by S1P_3_ receptor. These results taken together suggest that in clinical situations, when thrombotic coronary occlusion causes cardiac ischemia, the released S1P might compromise postischemic recovery due to its unfavorable coronary effects, which might outweigh the presumed cardioprotective effects of S1P produced by the ischemic myocardium. Clearly, further studies are warranted using in vivo and ex vivo models to obtain a better understanding of the (patho)physiological actions of S1P in ACS.

## Figures and Tables

**Figure 1 cells-09-01770-f001:**
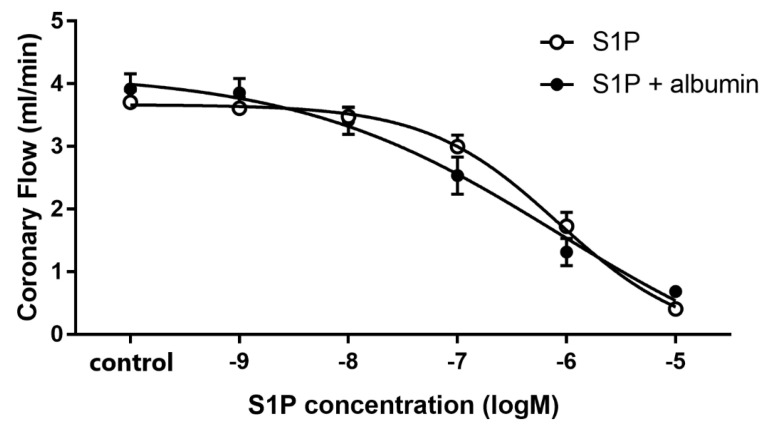
Dose-dependent effects of S1P on coronary flow of isolated murine hearts infused alone or in the presence of S1P-carrier albumin. In these experiments S1P was applied in a range of 10^−9^ to 10^−5^ M in cumulative doses without (S1P) or in the presence of its carrier, human serum albumin (S1P + albumin), and its effects on coronary flow were investigated. Albumin was present in a concentration twice that of S1P. ED_50_ values were 1.17 × 10^−6^ M (S1P) and 1.85 × 10^−7^ M (S1P + albumin). Mean ± SEM; *n* = 9; 8. Non-linear regression analysis and comparison of Fits using GraphPad Prism 7.0.

**Figure 2 cells-09-01770-f002:**
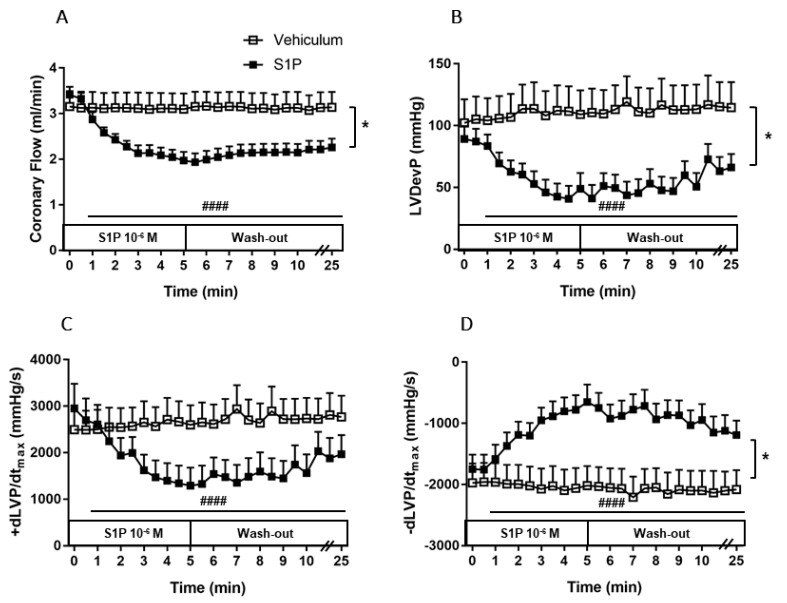
Effects of S1P on coronary flow (CF) (**A**), left ventricular developed pressure (LVDevP) (**B**), +dLVP/dt_max_ (**C**) and −dLVP/dt_max_ (**D**) of isolated mouse hearts. S1P (10^−6^ M) or its vehicle was administered to the perfusate of isolated wild-type (WT) murine hearts for 5 min. The infusion was followed by a 20-min wash-out period. Administration of S1P resulted in a remarkable decrease in CF, which prevailed throughout the infusion and the wash-out period (*p* < 0.0001). CF reduction compromised left ventricular contractile performance as evidenced by a concomitant decrease in LVDevP, +dLVP/dt_max_ and −dLVP/dt_max_ (*p* < 0.0001). Mean ± SEM; *n* = 6, 9; #### *p* < 0.0001 vs. baseline (pre-infusion value), * *p* < 0.05 vs. vehicle; two-way repeated measurement ANOVA and Dunnett’s *post hoc* test.

**Figure 3 cells-09-01770-f003:**
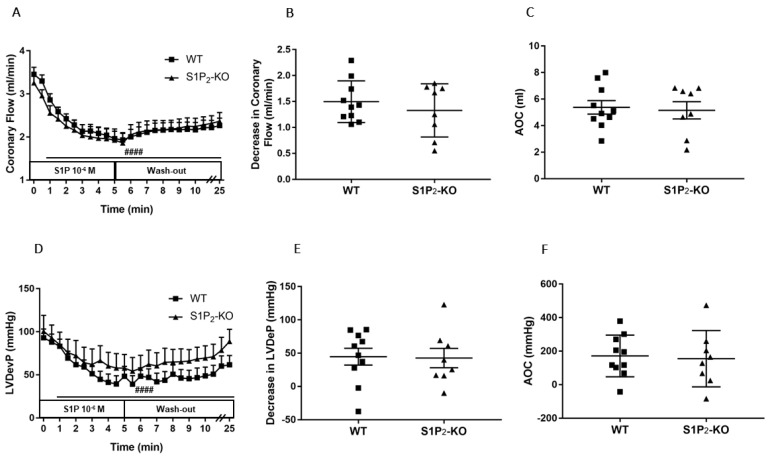
Effects of S1P on coronary flow (CF) (**A**–**C**) and left ventricular developed pressure (LVDevP) (**D**–**F**) of hearts isolated from wild-type (WT) and S1P_2_ knock-out (KO) mice. S1P (10^−6^ M) was administered to the perfusate of isolated WT and S1P_2_-KO murine hearts for 5 min. The infusion was followed by a 20-min wash-out period. CF and LVDevP were monitored during the entire experiment (panels A and D). Maximal decrease in CF and LVDevP compared to preinfusion baseline are shown in panels B and E. Values of area over the curve (AOC) during S1P infusion are shown in panels C and F. In S1P_2_-KO hearts S1P-induced CF and LVDevP reduction was similar to that observed in WT hearts. Mean ± SEM; *n* = 10, 8; #### *p* < 0.0001 vs. baseline (preinfusion value) in both groups, two-way repeated measurement ANOVA followed by Dunnett’s *post hoc* test.

**Figure 4 cells-09-01770-f004:**
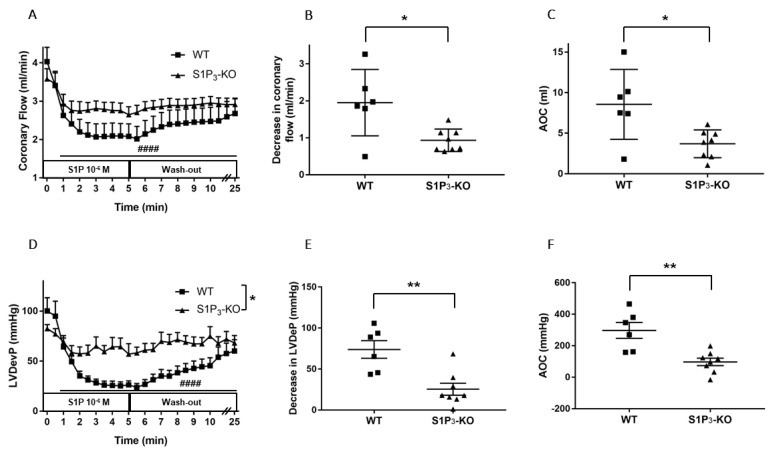
Effects of S1P on coronary flow (CF) (**A**–**C**) and left ventricular developed pressure (LVDevP) (**D**–**F**) of hearts isolated from wild-type (WT) and S1P_3_ knock-out (KO) mice. S1P (10^−6^ M) was administered to the perfusate of isolated WT and S1P_3_-KO murine hearts for 5 min. The infusion was followed by a 20-min wash-out period. CF and LVDevP are shown in panels A and D. Maximal decrease in CF and LVDevP compared to preinfusion baseline are shown in panels B and E. Values of area over the curve (AOC) during S1P infusion are shown in panels C and F. In S1P_3_-KO hearts, the S1P-induced CF and LVDevP reduction was significantly reduced. Mean ± SEM; *n* = 6, 8; #### *p* < 0.0001 vs. baseline (preinfusion value); * *p* < 0.05, ** *p* < 0.01 vs. WT; two-way repeated measurement ANOVA and Dunnett’s *post hoc* test (**A**,**D**) and unpaired *t*-test (**B**–**F**).

**Figure 5 cells-09-01770-f005:**
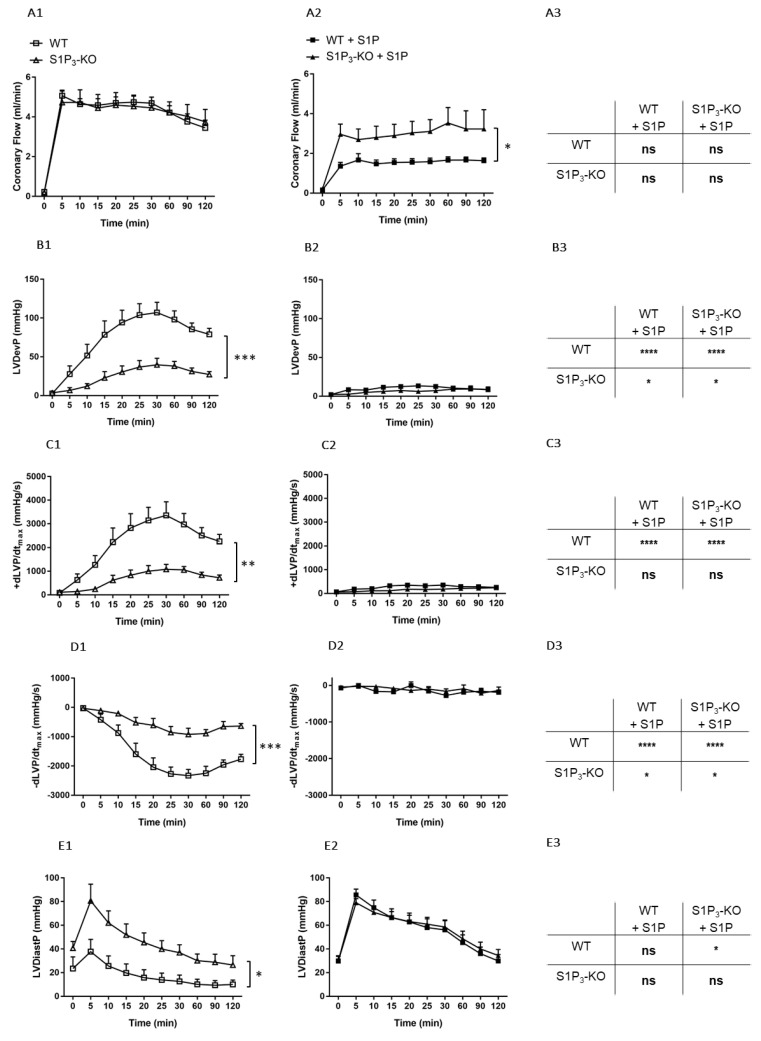
Postischemic coronary flow (CF) (**A**), left ventricular developed pressure (LVDevP) (**B**), +dLVP/dt_max_ (**C**), −dLVP/dt_max_ (**D**) and left ventricular diastolic pressure (LVDiastP) (**E**) in isolated WT and S1P_3_ knock-out (KO) mouse hearts without (left panels: **A1**–**E1**) or with S1P administration (middle panels: **A2**–**E2**) for 5 min to the perfusate at 10^−6^ M before the induction of a 20-min ischemia followed by a 120-min reperfusion period. The right panels (**A3**–**E3**) demonstrate statistical comparison of the parameters captured at the end of the reperfusion period. Mean ± SEM; *n* = 6, 8, 7, 7; * *p* < 0.05, ** *p* < 0.01, *** *p* < 0.001, **** *p* < 0.0001, with two-way repeated measurement ANOVA and Dunnett’s *post hoc* test in the graphs and two-way ANOVA followed by Sidak’s *post hoc* test in the table insets.

**Figure 6 cells-09-01770-f006:**
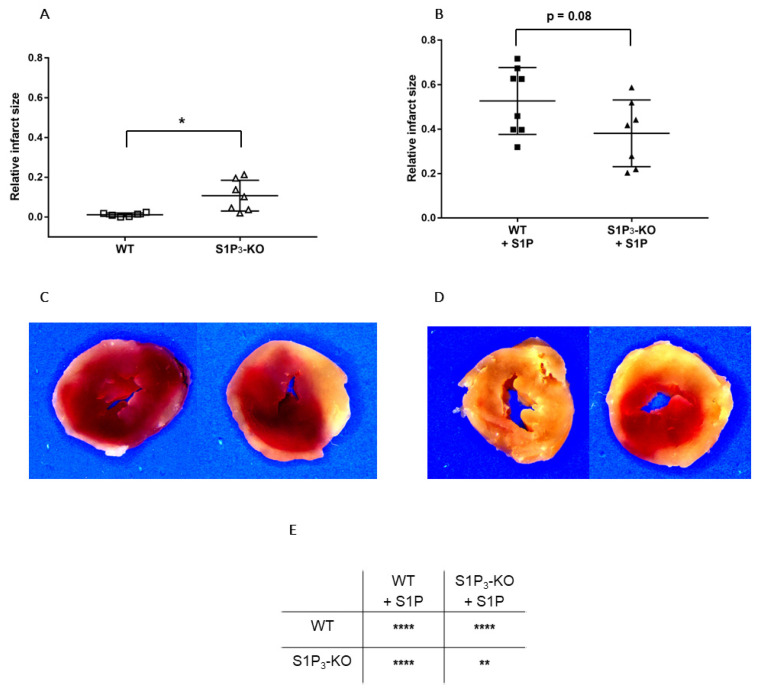
Relative infarct size (**A**,**B**) and representative sections (**C**,**D**) from hearts subjected to ischemia/reperfusion without (A & C) or with (B & D) 10^−6^ M S1P infusion. Mean ± SEM; *n* = 6, 8, 7, 7; * *p* < 0.05, ** *p* < 0.01, **** *p* < 0.0001, with unpaired *t*-test (**A**,**B**) or two-way ANOVA and Sidak’s multiple comparison test (**E**).

**Figure 7 cells-09-01770-f007:**
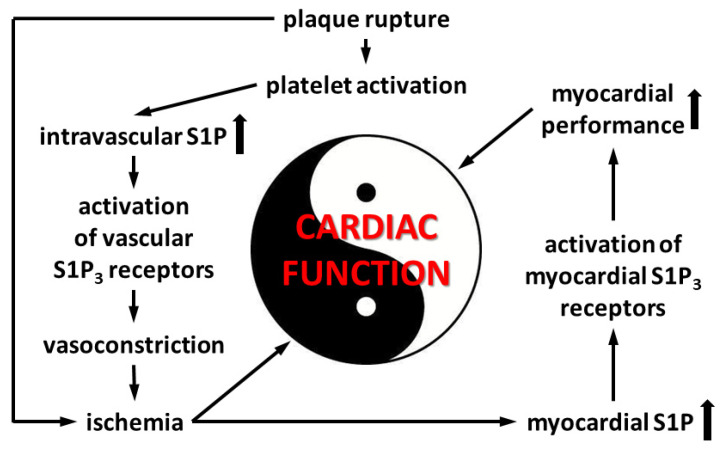
Events in acute coronary syndrome related to S1P_3_-mediated alterations of cardiac function.

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
