# Peer review of "Opposing Roles of S1P_3_ Receptors in Myocardial Function"

_cells, 2020, doi:10.3390/cells9081770_

Round 1

Reviewer 1 Report

This study by Wafa and co-workers has examined the effect of S1P receptors and S1P administration in the ex vivo Langendorf heart model. The authors have studied coronary blood flow and cardiac performance under normal conditions and after ischemia.

The authors have observed reduced coronary blood flow by S1P that was dependent on S1P3 leading to decreased cardiac function. In ischemia with subsequent reperfusion, S1P increased infarct size independently of S1P3 despite better CBF in the absence of S1P3.

One major problem with the data is statistics: in all figures where wt vs ko and +/- S1P is compared, a 2 way anova must be conducted. Not one way as the authors did. Much of the study cannot be assessed.

The second problem is that the authors found preischemic S1P treatment to increase infarct size, which is contrary to many studies in the literature where preischemic S1P treatment protects against ischemia reperfusion injury (in the heart and many other organs). Even if the authors argue that studies such as ref 22,23 showing the opposite were performed in vivo compared to the Landendorff perfused hearts ex vivo, there are plenty of other studies in Langendorff hearts showing a protective effect of preischemic S1P administration in the context of ischemia reperfusion injury; see reviews on the topic by Karliner et al (e.g. Karliner, J Cardiovasc Pharmacol. 2009 Mar;53(3):189-97. doi: 10.1097/FJC.0b013e3181926706. Sphingosine Kinase and Sphingosine 1-phosphate in Cardioprotection). Such discrepancies cannot be cast aside be the standard “difference in methods” phrase. Furthermore, studies in vivo showing beneficial S1P effects are physiologically far more significant than the contrary shown ex vivo as in the study here.

A further discrepancy is that contrary to the authors’ claim, other studies do not confirm their data: in fact, neither ref 22 nor 23 found larger infarct sizes in S1P3 KO mice as the authors did.

On line 210, the authors state they have investigated the role of myocardial S1P release. In fact, they have not. No data are provided that there is actually any S1P release from the myocardium.

I'd like to know what the p value is in 5B. It looks significant to me.

In summary, the coronary flow data appear interesting and agree with the literature on dogs and mice (S1P was shown to reduce myocardial perfusion though S1P3 in the murine heart, see High-density lipoprotein stimulates myocardial perfusion in vivo. Levkau B, Hermann S, Theilmeier G, van der Giet M, Chun J, Schober O, Schäfers M. Circulation. 2004 Nov 23;110(21):3355-9. doi: 10.1161/01.CIR.0000147827.43912.AE.).

The data on postischemic myocardial injury in relation to S1P and S1P3 are controversial without providing an explanation and contain flawed statistics.

Author Response

Please see the attachment for responses and also statistics files in the supplemetary material.

Reviewer 2 Report

In the manuscript ” Opposing roles of S1P3 receptors in myocardial function” authors have investigated the impact of Sphingosine-1-phosphate (S1P) on isolated perfused mice hearts on coronary flow and heart function under stable conditions and in the setting of IR-injury. To differential the effect through the S1P2-receptor and S1P3-receptor experiments were carried out on S1P2-KO, S1P3-KO, and wildtype mice. Authors found the under stable conditions S1P reduced coronary flow and reduces LVDP in wildtype and S1P2-KO mice, and to a less extent in S1P3-KO, suggesting that effects of S1P are predominantly mediated by the S1P3-receptor. Furthermore the authors found that S1P3-KO are more susceptible to IR-injury than wildtype mice with greater infarct size and worse cardiac function. Finally authors found that S1P administration in a very high dose prior to ischemia further increased infarct size, most predominate in wildtype, and a little less in S1P3-KO.

I have the following comments and concerns:

  1. The abstract does not follow the structure background, method, results, conclusion and needs to be revised to do this. In the current format it is difficult to get an overview of the method and thereby to understand how you came to the results in the experiment.
  2. Line 36: please revise – ex.: This process results in thrombotic occlusion …
  3. Line 114-121: The experimental protocol is very poorly explained. Part of the protocol description is misplaced in the results section. The method should be explained in the method section and the protocol should be clear there. Please revise this part of the method section. Preferably add an illustration of the perfusion protocols.
  4. Figure 5A: Why is difference in infarct size in this figure presented with a bar and *, when this no nomenclature is not used in figure 5B and already illustrated in 5E? Please remove.
  5. Line 263-264: Please delete consequential, as you can not conclude this. If you mean consequential in terms of due to – this can not be concluded by your experiments. All you can conclude is that reduction of coronary flow is simultaneous with reduction in LVDP, not that there is causality. It you want to conclude on causality you should also provide experiment with experiments groups with langendorf perfusion with constant flow instead of constant pressure. Please revise.
  6. Line 267 “ enhanced suppression” is confusing – please rephrase.
  7. Line 274: Authors write that knocking out S1P3 failed to improve cardiac function or to reduce infarct size, while according to figure 5E there was significant smaller infarct size in S1P3-KO that in wildtype. Please rephrase.
  8. Line 275-277. Please rephrase.
  9. Line 288-290. Please add reference to substantiate that 1 microM may mimic ACS
  10. Several passages in the discussion authors state that they mimic an ACS in their experiment. It is fair to explain that S1P is released in the coronary arteries in ACS and this is the reason for the protocol, but further similarity is spares. Therefor please tone this statement down in the manuscript. And please provide reference when comparison is made (ex. line 356-358)
  11. Line 305: Authors conclude that S1P3 plays the most import role in mediating S1p-incuded CF reduction. Please rephrase. Since only 2 receptors are investigated in the study it can not be conclude from this study that it is the most important in general.
  12. Authors conclude that S1P has both protective and detrimental effect and these outweigh each other. But 1 microM S1P is compared with other studies a very high dose, and in this study an obvious toxic dose as both wild type and S1P3-ko had markedly increased infarct size and no cardiac function. Please provide dose-response experiment results. The aim of the study is to show the harmful and protective effects of S1P3. As such the IR-protocol should be set up so the control (wildtype) mice have infarct size of about 50%, so any protective as well as harmful effect will be evident in the experiment.

Reviewer 3 Report

In this manuscript, the authors investigated the role of S1P receptors (especially S1P3) in myocardial function using Langendorff-perfused murine hearts. It was demonstrated that S1P reduced coronary flow and contractile performance via S1P3 (not S1P2), and that S1P3 plays a protective role in post-ischemic functional recovery, suggesting that S1P3 seems to mediate two opposing (deleterious and protective) actions in the heart.

I think that the article is intriguing and worth publishing in “Cells”, although it could be improved if the data for S1P2-KO in the experimental settings of Figure 4 are presented.

Round 2

Reviewer 1 Report

Issues habe been clarified. Accept as it is.